# Bone Marrow Adipocytes Contribute to Tumor Microenvironment-Driven Chemoresistance via Sequestration of Doxorubicin

**DOI:** 10.3390/cancers15102737

**Published:** 2023-05-12

**Authors:** Jun-Goo Kwak, Jungwoo Lee

**Affiliations:** 1Molecular and Cellular Biology Graduate Program, University of Massachusetts Amherst, Amherst, MA 01003, USA; jungookwak@umass.edu; 2Department of Chemical Engineering, University of Massachusetts Amherst, Amherst, MA 01003, USA; 3Institute for Applied Life Sciences, University of Massachusetts Amherst, Amherst, MA 01003, USA

**Keywords:** bone marrow adipocyte, 3D cell culture, chemoresistance, tumor microenvironment, multiple myeloma

## Abstract

**Simple Summary:**

Despite substantial progress and effort, the clinical efficacy of chemotherapy remains suboptimal due to chemoresistance, which is partly attributed to adipocytes in the tumor microenvironment. Adipocytes, major stromal cells in the bone marrow, are known to uptake lipophilic molecules. As many chemotherapeutics are hydrophobic, bone marrow adipocytes may contribute to the development of chemoresistance. However, detailed investigation has been challenging due to the limited accessibility and heterogeneity of the bone marrow microenvironment. To address these barriers, we have established 2D and 3D in vitro culture models of human bone marrow adipocytes. Our models provide compelling evidence that bone marrow adipocytes sequester doxorubicin in lipid droplets, resulting in a reduction in its cytotoxicity. These findings reveal an orthogonal mechanism for bone marrow tumor microenvironment-driven chemoresistance.

**Abstract:**

Chemoresistance is a significant problem in the effective treatment of bone metastasis. Adipocytes are a major stromal cell type in the bone marrow and may play a crucial role in developing microenvironment-driven chemoresistance. However, detailed investigation remains challenging due to the anatomical inaccessibility and intrinsic tissue complexity of the bone marrow microenvironment. In this study, we developed 2D and 3D in vitro models of bone marrow adipocytes to examine the mechanisms underlying adipocyte-induced chemoresistance. We first established a protocol for the rapid and robust differentiation of human bone marrow stromal cells (hBMSCs) into mature adipocytes in 2D tissue culture plastic using rosiglitazone (10 μM), a PPARγ agonist. Next, we created a 3D adipocyte culture model by inducing aggregation of hBMSCs and adipogenesis to create adipocyte spheroids in porous hydrogel scaffolds that mimic bone marrow sinusoids. Simulated chemotherapy treatment with doxorubicin (2.5 μM) demonstrated that mature adipocytes sequester doxorubicin in lipid droplets, resulting in reduced cytotoxicity. Lastly, we performed direct coculture of human multiple myeloma cells (MM1.S) with the established 3D adipocyte model in the presence of doxorubicin. This resulted in significantly accelerated multiple myeloma proliferation following doxorubicin treatment. Our findings suggest that the sequestration of hydrophobic chemotherapeutics by mature adipocytes represents a potent mechanism of bone marrow microenvironment-driven chemoresistance.

## 1. Introduction

Despite significant advancements in chemotherapeutic agents and regimens, chemotherapy often fails because disseminated tumor cells develop chemoresistance in peripheral tissues [1,2]. Previous studies have revealed that genetic mutations and alterations in cellular signaling pathways during chemotherapy promote the survival and fitness of a subset of tumor cells [3,4,5]. Accumulating evidence has also demonstrated that tumor microenvironments, where tumor cells interact with tissue-resident stromal cells, extracellular matrix proteins, and soluble factors, play a major role in developing chemoresistance by establishing a protective niche for disseminated tumor cells. Therefore, understanding how tissue-specific tumor microenvironments contribute to chemoresistance is imperative to improving the efficacy of chemotherapy for long-term metastasis prevention [6,7,8,9].

Bone is a major site for metastasis, along with the liver, lung, and brain [10,11]. Bone metastasis begins with tumor cell dissemination to the bone marrow, a vascularized semi-solid tissue inside bone cavities. Within the bone marrow, adipocytes are a major stromal component, taking up over 50% of the bone marrow volume at the time of adult cancer diagnosis [12,13,14]. Previous studies have identified a general link between adipocytes and tumor microenvironments that alters the clinical outcome of chemotherapy. For instance, obese patients require higher doses of chemotherapeutic agents as they show reduced efficacy compared to patients with a lower body mass index. This is partly because adipose tissue can affect the uptake or efflux of chemotherapeutic agents, which compromises chemotherapy efficacy [15,16]. In addition, adipocytes have been shown to secrete proinflammatory molecules and provide fatty acids, which can promote the survival and growth of tumor cells in local tissue microenvironments [17,18,19,20,21,22]. However, the exact role of adipocytes in modulating the efficacy of chemotherapy in the context of the bone marrow remains uncertain, mainly due to the anatomical inaccessibility of bone marrow cavities and the limited ability to investigate their specific contribution in a highly complex and heterogeneous microenvironment.

In vitro differentiated human bone marrow adipocyte culture models have been developed as an alternative approach for studying the impact of adipocytes on chemotherapy efficacy [23,24,25,26]. These models allow the creation of controlled tissue microenvironments and a detailed investigation of the interactions between adipocytes and chemotherapeutic agents. For instance, in vitro models enable the creation of adipocytes with different properties, such as increased or decreased lipid content, which can alter drug uptake and clearance [27]. In vitro models further support high-throughput experiments, expediting the identification of chemotherapeutic agents or combination therapies that are more effective against cancer cells and can be applied to identify regimens that have increased efficacy in the presence of adipocytes [28,29]. Additionally, 3D adipocyte culture models enable the investigation of cell-cell interactions, diffusion limits, and densified cultures that are difficult to recapitulate in 2D plastic culture but are present in native tissues [30,31]. These studies have demonstrated the importance of adipocytes in tumor microenvironments but also highlight the need for standardized and accelerated protocols to shorten the required time for mature adipocyte development in a reproducible manner.

In this paper, we report preclinical evidence that in vitro differentiated bone marrow adipocytes contribute to tumor microenvironment-driven chemoresistance and develop 2D and 3D human bone marrow adipocyte culture models. We first established an accelerated adipogenic differentiation protocol of human bone marrow stromal cells (hBMSCs) by introducing rosiglitazone, an agonist of peroxisome proliferator-activated receptor gamma (PPARγ), which is a key regulator of adipogenesis [32]. Next, we prepared a 3D in vitro differentiated bone marrow adipocyte model using an inverted colloidal crystal (ICC) hydrogel scaffold that mimics the sinusoid-relevant architecture and biophysical environment found in bone marrow [33,34,35,36,37]. hBMSCs introduced in this 3D hydrogel scaffold formed spherical aggregates and underwent robust differentiation into adipocytes with rosiglitazone. The established 3D adipocyte culture model substantiated the sequestration of lipophilic doxorubicin and the functional consequences of modulating cytotoxicity in hBMSCs through a conditioned medium experiment. Finally, we recapitulated chemotherapy in a bone marrow model of multiple myeloma by directly coculturing 3D in vitro differentiated human bone marrow adipocytes and human multiple myeloma cells (MM1.S) in the presence of doxorubicin [38].

## 2. Materials and Methods

Chemicals and cell culture reagents were purchased from Fisher Scientific, unless stated otherwise.

### 2.1. Inverse Colloidal Crystal Hydrogel Scaffold Fabrication

First, soda lime glass beads (diameter = 300 ± 16 μm) were resuspended in deionized water and introduced into 1 mL borosilicate glass vials (inner diameter = 6.5 mm). Beads were added in a dropwise manner to a final height of ~2 mm. Next, the glass vials with glass beads were sonicated in an ultrasonic water bath for 30 min and placed in a vacuum chamber for 4 h to promote orderly packing and remove entrapped air bubbles, respectively. After complete evaporation of water at 65 °C in an oven overnight, the glass beads were sintered at 674 °C for 5 h to partially fuse the glass beads into a single structure. After sintering, these free-standing colloidal crystal templates were transferred to a slightly larger vial (inner diameter = 8 mm) to facilitate extra void space on each face of the colloidal crystal templates prior to polymer precursor solution infiltration.

A polyacrylamide precursor solution was prepared using 30% (*w*/*v*) acrylamide (monomer) and *N*,*N′*-methylenebisacrylamide (crosslinker) (19:1 ratio) in nitrogen gas-purged Milli-Q water. Darocur 1173 (photoinitiator, 0.2% (*v*/*v*)) and tetramethylethylenediamine (catalyst, 0.2% (*v*/*v*)) were subsequently added and thoroughly mixed with a vortex. The complete hydrogel precursor solution (200 μL) was infiltrated into each colloidal crystal template by centrifugation at 10,000 r.p.m. for 15 min. Polymerization was then initiated under a 15 W UV lamp for 15 min and left overnight to ensure reaction completion. After polymerization, the hydrogel-infiltrated colloidal crystals were extracted from the glass vials, and only the bottom face of the polymer was removed from the template using a razor blade to make a pocket-type ICC hydrogel scaffold [37,39].

Next, the entrapped glass beads were dissolved through a sequential immersion of 10% (*v*/*v*) hydrofluoric acid solution in 2.4 M hydrochloric acid solution for 2 h and then a subsequent wash in 1.2 M hydrochloric acid solution for 2 h under orbital shaking at 100 r.p.m. This sequential solution cycle was repeated until no further glass beads remained, as determined by visual optical transparency. Upon dissolution of the glass beads, the scaffolds were washed 5 times in Milli-Q water, frozen at −80 °C, and lyophilized overnight to remove excess solvent and acid. After freeze drying, the scaffolds were then rehydrated 3 times in sterile PBS and centrifuged at 1500 r.p.m. for 5 min to remove air bubbles entrapped in the hydrogel scaffold cavities. Prior to cell seeding, the hydrogel ICC scaffolds were sterilized in 70% ethanol under a 15 W UV light for 30 min, stabilized in sterile PBS 3 times, and centrifuged for 5 min at 1500 r.p.m.

### 2.2. Adipocyte Differentiation of Human Bone Marrow Stromal Cells

Fresh human bone marrow aspirate was obtained from healthy donors (STEMCell Technologies, Vancouver, BC, Canada). First, bone marrow was diluted in a 1:1 ratio with sterile PBS, layered over Ficoll Paque (Cytiva, Marlborough, MA, USA) density gradient solution, centrifuged at 400× *g* for 30 min using minimum acceleration without the brake, and the bone marrow mononuclear cells isolated from the upper layer. Next, bone marrow mononuclear cells were plated on tissue culture flasks and cultured with Minimum Essential Medium alpha (α-MEM) supplemented with the following: 10% (*v*/*v*) fetal bovine serum (FBS), 1% (*v*/*v*) penicillin-streptomycin (PS), 20 mg/L gentamycin, and 1 μg/L recombinant human fibroblast growth factor. After 1 week, colony-forming human bone marrow stromal cells (hBMSCs) were collected, replated for 3 weeks for expansion, and cryopreserved (passage 0). hBMSCs with a passage > 4 were used in experiments.

Next, we prepared two adipogenic mediums with different compositions: (i) the first adipogenic medium (AM1) was composed of high-glucose DMEM supplemented with the following: 10% (*v*/*v*) FBS, 1% (*v*/*v*) PS, 1 μM dexamethasone, 500 μM 3-isobutyl-1-methylxanthine, 10 μM rosiglitazone, and 10 μg/mL of insulin from bovine pancreas; and (ii) the second adipogenic medium (AM2) was composed of high-glucose DMEM supplemented with 10% (*v*/*v*) FBS, 1% (*v*/*v*) PS, and 10 μg/mL of insulin from bovine pancreas. After seeding hBMSCs in 96-well plates (2D) or scaffolds (3D), the culture was initially maintained in α-MEM until hBMSCs reached 80% confluence in 96-well plates or 1 week of culture in ICC hydrogel scaffolds. hBMSCs were then differentiated into adipocytes by culturing with AM1 for 9 days to induce adipogenesis and then AM2 for an additional 9 days to facilitate lipid droplet expansion while changing medium every 3 days.

### 2.3. Cell Seeding of ICC Scaffolds for Spheroid Formation

Sterilized ICC hydrogel scaffolds were placed in a 24-well plate, partially dehydrated in a biosafety cabinet for 20 min, and extracted from each scaffold. Subsequently, 20 μL of cell suspension at a density of 20,000 cells/μL was inoculated on top of the dehydrated scaffold to seed 400,000 total hBMSCs by capillary action. Next, the seeded scaffolds were placed in a 37 °C tissue culture incubator with 5% CO_2_ for 30 min prior to the addition of 1 mL of expansion media to promote cell aggregation and spheroid formation. After 24 h, the scaffolds were placed into a new 24-well plate due to residual hBMSCs proliferating in the well post-seeding and then cultured with adipogenic differentiation medium to create adipocyte spheroids (9 days with AM1 followed by 9 days with AM2).

### 2.4. Actin, Lipid, and Nucleus Staining

hBMSCs and adipocytes in 2D and 3D cultures were first fixed with 4% (*w*/*v*) paraformaldehyde for 5 min and washed 3 times with sterile PBS. Next, fixed cells were permeabilized with a 0.05% (*v*/*v*) Tween-20 solution for 10 min and washed 3 times with sterile PBS. Neutral lipid droplets retained in adipocytes after permeabilization were stained with 10 ng/mL of BODIPY-FL dye for 30 min. After 3 washes with sterile PBS to remove excess BODIPY-FL (green) dye, cells were incubated with phalloidin-TRITC (red) and DAPI (blue) solutions for 1 h to stain filamentous actin and nuclei. After labeling the actin fibers and nuclei, the excess staining solution was washed away with sterile PBS.

### 2.5. Fluorescence Microscopy Imaging

Stained hBMSCs and adipocytes in a 96-well plate were imaged with an EVOS M7000 fluorescence microscope with DAPI, GFP, and RFP filter cubes as well as 4×, 10×, and 20× objectives. Fluorescence images obtained from the EVOS M7000 were processed in ImageJ.

### 2.6. Confocal Microscopy Imaging

Stained hBMSCs and adipocytes in ICC hydrogel scaffolds were placed in a 35-mm Petri dish with PBS. Samples were then imaged with a Nikon A1R25 confocal microscope with a 10× Apo objective using 405, 488, and 561 laser lines. After the acquisition, images were exported and then processed in Nikon NIS-Elements AR v4.53.

### 2.7. Doxorubicin Sequestration Assay

hBMSCs were seeded in a 96-well plate (2D) or in scaffolds (3D) and differentiated into adipocytes for 18 days following the previously described adipogenic differentiation protocol. hBMSCs were passage-matched for all experiments. PBS control or 2.5 μM doxorubicin (Tocris Bioscience, Bristol, UK) was introduced to hBMSCs and bone marrow adipocytes for 3 days. Next, doxorubicin fluorescence was imaged with an EVOS M7000 fluorescence microscope to determine the extent of doxorubicin sequestration. To measure doxorubicin release, media was changed every 3 days for 1 week while measuring fluorescent intensity prior to each medium change. To quantitatively calculate doxorubicin sequestration and release as a function of fluorescence intensity, the following formula was used: Corrected total cell fluorescence (CTCF) = Integrated density − (Area of selected cell × Mean fluorescence of background readings).

### 2.8. Doxorubicin Cytotoxicity Assay in Indirect Culture with Conditioned Medium

hBMSCs and adipocyte scaffolds were prepared as previously described. Doxorubicin (2.5 μM) or PBS control was introduced for 72 h, after which conditioned media were collected. In separately prepared 96-well plates, hBMSCs were stained with 20 mM live nuclear staining solution (DAPI, Hoechst 33342) for 30 min to determine the initial total number of nuclei. These naïve hBMSCs were then incubated with the conditioned media from hBMSCs or adipocyte scaffolds containing doxorubicin or PBS control. After 3 days of culture, cells were imaged once more to quantify the remaining nuclei and the changed cell morphology. Cytotoxicity was quantitatively measured by the decreased number of nuclei after exposure to the conditioned medium.

### 2.9. Doxorubicin Cytotoxicity Assay in Direct Coculture with Multiple Myeloma Cells

hBMSCs and adipocyte scaffolds were prepared as previously described, punched using a 5 mm biopsy punch, and placed into 24-Transwell inserts with 0.38 μm pores. Next, 5000 eGFP-MM1.S multiple myeloma cells were seeded into each scaffold with doxorubicin (2.5 μM) or PBS control. Fluorescence microscopy was performed with an EVOS M7000 microscope, and images were acquired every day for 4 days to determine multiple myeloma cell proliferation kinetics. Cytotoxicity was quantitatively determined by measuring the surface area covered by eGFP-MM1.S multiple myeloma cells based on the control.

### 2.10. Statistical Analysis

All data points were used without pre-processing for statistical analysis using GraphPad Prism 9. Unpaired Student’s *t*-test was performed when comparing the mean values between the two groups. For comparisons between more than two groups, a two-way ANOVA was performed with Bonferroni’s post hoc analysis with α = 0.05, as appropriate. All quantitative data represent the mean ± standard error of the mean with * *p* < 0.05, ** *p* < 0.01, and *** *p* < 0.001.

## 3. Results

### 3.1. Rosiglitazone Promotes Rapid Adipogenic Differentiation of hBMSCs

PPARγ is a nuclear receptor that plays a crucial role in promoting adipogenesis via modulating glucose and lipid metabolism [40]. However, chronic activation of PPARγ has also been shown to decrease mature lipid formation [41]. We hypothesized that introducing rosiglitazone, a PPARγ agonist, during the early stage of adipogenesis could significantly accelerate mature adipocyte differentiation of hBMSCs by enhancing adipogenic lineage commitment of hBMSCs and thereby accelerate mature adipocyte differentiation. To test this hypothesis, we established primary hBMSCs from a healthy donor’s bone marrow aspirate and induced adipogenic differentiation of hBMSCs by adding rosiglitazone to conventional adipogenic differentiation medium containing dexamethasone, IBMX, and insulin for 9 days [42,43,44]. Next, we cultured these preadipocytes in an adipogenic maintenance medium supplemented only with insulin for 9 additional days to increase lipid droplet hypertrophy. As a control, hBMSCs were differentiated with the same two-stage differentiation protocol without rosiglitazone (Figure 1a). On Days 6, 12, and 18, we characterized adipocyte morphology and differentiation through fluorescent actin and lipid staining using phalloidin and BODIPY dyes. Time-course fluorescent images confirmed significantly accelerated adipogenesis with rosiglitazone (Figure 1b).

To determine the extent of adipogenesis, we measured time-course changes in the diameter of lipid droplets within adipocytes. Lipid droplets became evident on Day 6, in which hBMSCs differentiated with rosiglitazone had significantly larger lipid droplet diameter (5.66 ± 0.07 μm) compared to those without rosiglitazone (3.41 ± 0.08 μm). These significant differences were maintained on Day 12. At the end of 18 days of differentiation, adipocyte lipid droplet diameter with rosiglitazone (18.4 ± 0.31 μm) was 1.8-fold larger than control (10.5 ± 0.13 μm). Characterization of the kinetics of lipid droplet formation further substantiated the significantly accelerated adipogenesis of hBMSCs with rosiglitazone (1.11 μm/day) compared to without rosiglitazone (0.56 μm/day) (Figure 1c).

We measured the surface area covered by lipid droplets with BODIPY, a nonpolar dye that binds to neutral lipid droplets. The BODIPY signal was thresholded and normalized to the entire surface area of the cells. Quantitative image analysis of lipid surface area by lipophilic BODIPY dye additionally supported significantly accelerated adipogenesis with rosiglitazone from Day 6 (mean surface area was 2.88 ± 0.33% without rosiglitazone vs. 9.00 ± 0.65% with rosiglitazone). By Day 18 of culture, ~37% of the surface area was stained with BODIPY dye in adipocytes differentiated with rosiglitazone, whereas the lipid surface area was only ~13% in the control differentiation medium. Characterization of the kinetics of BODIPY surface area confirmed that hBMSC differentiated with rosiglitazone was nearly 3-fold faster (2.33%/day) in comparison to the control (0.82%/day) (Figure 1d). Collectively, these results support the fact that the addition of rosiglitazone significantly accelerates adipogenesis and mature lipid droplet formation compared to conventional adipogenic differentiation medium.

### 3.2. Development of a 3D Bone Marrow Adipocyte Model with ICC Hydrogel Scaffolds

We next applied the established adipogenic differentiation protocol to a 3D spheroid culture model using an inverse colloidal crystal (ICC) hydrogel scaffold that mimics the sinusoid anatomy and biophysical complexity of the bone marrow in a controlled and reproducible manner [45,46]. Colloidal crystal templates with a diameter of 6.5 mm and a height of 2 mm were prepared by densely packing glass beads (diameter = 300 ± 16 μm) and thermally sintering them. We then infiltrated the templates with a polyacrylamide hydrogel precursor solution and, after polymerization, selectively dissolved the glass beads, leaving behind an ICC porous hydrogel scaffold. Optical microscopy confirmed the presence of defined pores with interconnecting junctions, representing ~15% of the cavity diameter (Figure 2a).

A pocket-shaped ICC scaffold with open pores only on the top side allowed for over 90% seeding efficiency of hBMSCs. The spherical pore cavities guided cellular aggregation and spheroid formation [37]. Subsequent cultures of 3D hBMSC spheroids in adipogenic differentiation medium with and without rosiglitazone induced adipogenesis (Figure 2b). Confocal microscopy imaging of the 3D spheroid culture after BODIPY staining at Days 6, 12, and 18 revealed the formation of lipid droplets (Figure 2c).

We conducted a quantitative analysis to track changes in the diameters of 3D hBMSC spheroids in pore cavities over time. In adipogenic medium without rosiglitazone, 3D spheroids maintained comparable diameters throughout the entire 18-day period of differentiation (38.2 ± 2.5 μm). In contrast, the size of 3D spheroids cultured in adipogenic differentiation medium with rosiglitazone grew consistently larger than control starting on Day 12 (55.8 ± 2.2 μm) and Day 18 (68.0 ± 2.1 μm). Characterization of the kinetics of spheroid diameter confirmed that rosiglitazone addition had a significant effect with a much faster diameter rate of change (1.82 μm/day) compared to control without rosiglitazone, where the diameter rate of change remained stable (−0.075 μm/day) (Figure 2d).

We also conducted a comparative analysis of time-course changes in lipid droplet sizes in 3D adipocyte spheroids with and without rosiglitazone. Adipogenic differentiation medium including rosiglitazone significantly increased lipid droplet diameter (3.34 ± 0.14 μm) compared to the control medium without rosiglitazone (2.56 ± 0.17 μm) starting from Day 12. This trend continued to Day 18 (5.71 ± 0.21 μm with rosiglitazone vs. 3.04 ± 0.19 μm without rosiglitazone). The characterized rate of lipid droplet diameter change with rosiglitazone (0.256 μm/day) was ~4-fold faster than without rosiglitazone (0.058 μm/day) (Figure 2e). These results collectively indicate that adipogenic differentiation medium with the addition of rosiglitazone effectively promoted adipogenesis of hBMSCs in 3D spheroid culture, similar to adipogenesis in 2D hBMSC culture.

### 3.3. Bone Marrow Adipocytes Sequester the Anthracycline Chemotherapeutic Doxorubicin in Both 2D and 3D

We investigated the role of in vitro differentiated bone marrow adipocytes in developing chemoresistance using the established 2D and 3D bone marrow adipocyte models with doxorubicin. Doxorubicin is a common chemotherapeutic agent known for its high hydrophobicity and strong intrinsic fluorescence. Under the hypothesis that in vitro differentiated bone marrow adipocytes would sequester doxorubicin, resulting in a decrease in its therapeutic efficacy, we incubated hBMSCs and adipocytes in 2D TCP with 2.5 μM doxorubicin for 3 days (Figure 3a). Fluorescence microscopy revealed significantly higher fluorescence intensity inside adipocytes compared to hBMSCs. Corrected total cell fluorescence (CTCF) quantitatively corroborated significant sequestration of doxorubicin by in vitro differentiated bone marrow adipocytes in 2D TCP (CTCF: 7.21 ± 0.62) compared to hBMSC control (CTCF: 1.0 ± 0.16) (Figure 3b).

We repeated this experiment in 3D hBMSC and adipocyte spheroid cultures. After 3 days of incubation with 2.5 μM doxorubicin, fluorescence microscopy imaging confirmed the sequestration of doxorubicin by bone marrow adipocyte spheroids. CTCF in 3D adipocyte spheroids (CTCF: 6.27 ± 0.47) was similar to that in bone marrow adipocytes in 2D TCP (Figure 3c).

Next, we performed comparative longitudinal monitoring of sequestered doxorubicin in 2D and 3D adipocytes for 6 days under fluorescence microscopy. Quantification of acquired images revealed that fluorescence intensity decreased over time in both 2D and 3D adipocyte cultures. Interestingly, fluorescence intensity dropped significantly in 3D adipocyte spheroids on Day 6, while the 2D adipocytes still retained substantial fluorescence (Figure 3d). Quantitative analysis of CTCF kinetics confirmed not only decreasing doxorubicin fluorescence intensity in both 2D and 3D adipocyte cultures over 6 days but also significantly different rates of change. As a result, the CTCF at Day 6 in 3D adipocyte spheroids (0.103 ± 0.0023) was about 3-fold less than in the 2D adipocytes (0.331 ± 0.027) after 1 week (Figure 3e). Characterization of the rate of change in CTCF (CTCF/day) in 2D bone marrow adipocytes was −0.111/day, which was significantly lower compared to 3D adipocyte spheroids at −0.150/day. Collectively, these results support the sequestration of doxorubicin by 2D and 3D bone marrow adipocyte culture models. Direct longitudinal monitoring of doxorubicin with microscopy revealed decreasing fluorescence intensity over time after sequestration. Given that there were different kinetics of decreasing fluorescence intensity, this potentially implies distinct cellular metabolism between 2D and 3D bone marrow adipocyte cultures.

### 3.4. Sequestration of Doxorubicin by Bone Marrow Adipocytes Reduces Its Cytotoxicity in Conditioned Medium

Next, we investigated the impact of doxorubicin sequestration by in vitro differentiated bone marrow adipocytes on modulating chemotherapy efficacy as measured by cytotoxicity. Under the hypothesis that the sequestration of doxorubicin by in vitro differentiated bone marrow adipocytes would reduce its concentration in the surrounding medium, we tested this by quantifying doxorubicin’s cytotoxic effects on naïve hBMSCs after incubating with hBMSC or adipocyte spheroids first (Figure 4a). We measured the cytotoxicity of doxorubicin on naïve hBMSCs by staining with DAPI (Hoechst 33342), a cell-permeable nuclear dye that can be applied to live cells. After nuclei staining, these naïve hBMSCs were then incubated with conditioned media from hBMSC or adipocyte spheroids that were treated with or without doxorubicin for 72 h. Fluorescence microscopy imaging of hBMSCs stained with DAPI confirmed significant cytotoxicity and apoptosis, as indicated by a reduction in the total number of nuclei after incubation with the conditioned medium (Figure 4b).

Quantification of the relative fold-change in nuclei confirmed a highly significant reduction in the number of naïve hBMSCs (0.321 ± 0.029) when treated with conditioned media from hBMSC spheroids incubated with doxorubicin (Figure 4c). Critically, when naïve hBMSCs were incubated with conditioned media from adipocyte spheroids containing doxorubicin, there was no significant difference in the fold-change of nuclei compared to control (Figure 4d). Based on these results, we concluded that adipocyte sequestration of doxorubicin reduced its therapeutic efficacy in the conditioned medium through indirect culture.

### 3.5. Sequestration of Doxorubicin by 3D Bone Marrow Adipocytes Reduces Its Cytotoxicity in Direct Coculture with Human Multiple Myeloma Cells

We finally examined if sequestration of doxorubicin by 3D in vitro differentiated human bone marrow adipocytes reduces its cytotoxicity in the context of direct coculture with human cancer cells. A genetically labeled multiple myeloma cell line, eGPF-MM1.S, was used as a model cancer cell. We established a coculture system by introducing 5000 eGFP-MM1.S cells into a 24-well plate with 3D hBMSC or adipocyte scaffolds placed in a Transwell insert to prevent direct cell-to-cell contact while permitting medium exchange with a semi-permeable membrane. Next, 2.5 μM doxorubicin was introduced into the Transwell insert, and fluorescence microscopy imaging was performed daily for 4 days to assess the impact of chemotherapeutic sequestration on multiple myeloma cell growth (Figure 5a).

After the addition of doxorubicin, fluorescence images of eGFP-MM1.S cells showed a decrease in the surface area covered by eGFP-MM1.S cells in direct coculture (Figure 5b). Quantitative analysis of eGFP-MM1.S revealed significant differences in proliferation post-chemotherapy when cocultured with adipocyte spheroids on Day 2 (eGFP-MM1.S area: 0.532 ± 0.026 with hBMSC spheroids vs. 0.702 ± 0.031 with adipocyte spheroids) and on Day 4 (eGFP-MM1.S area: 0.591 ± 0.019 with hBMSC spheroids vs. 0.768 ± 0.032 with adipocyte spheroids) compared to control (Figure 5c). Taken together, these results illustrate how the sequestration of doxorubicin by in vitro differentiated bone marrow adipocytes reduces its cytotoxicity and how the bone marrow microenvironment reduces chemotherapeutic efficacy as a function of adipocyte content in the context of cancer.

## 4. Discussion

The objective of this study was to determine the impact of bone marrow adipocytes on the development of chemoresistance by establishing 2D and 3D human bone marrow adipocyte culture models. Adipocytes, a major stromal constituent in the bone marrow, tend to expand during aging, obesity, and osteoporosis, which negatively effects hematopoiesis and osteogenesis [47,48,49,50]. The accumulation of bone marrow adipocytes may also compromise the effectiveness of chemotherapy by modulating the tumor microenvironment. Previous studies have identified three distinct processes through which peripheral adipocytes modulate cancer treatment. First, adipocytes can support the survival and growth of tumor cells by secreting various cytokines, adipokines, and growth factors, such as IL-6 and leptin [51,52]. Second, adipocytes can facilitate tumor cell growth and metastasis through direct contact, changing sensitivity to chemotherapy in addition to undergoing metabolic reprogramming [53,54]. Third, adipocytes can alter the biodistribution of chemotherapeutic agents by sequestering and metabolizing lipophilic molecules, which can compromise the efficacy of chemotherapy in vivo [20].

A detailed investigation of the role of bone marrow adipocytes in impeding chemotherapy efficacy in mouse models has been challenging due to restricted anatomical accessibility and the highly heterogeneous nature of the bone marrow microenvironment. To address these challenges, we have developed in vitro models to provide well-defined and accessible platforms for studying the role of bone marrow adipocytes in developing chemoresistance. One critical barrier for in vitro culture of bone marrow adipocytes is to expedite and enhance the adipogenesis of hBMSCs, which can also differentiate into osteogenic cells [55,56]. Supplementation with dexamethasone, IBMX, and indomethacin can induce adipogenic differentiation of hBMSCs; however, developing a mature adipocyte phenotype takes longer than four weeks. Moreover, during this period, a subset of hBMSCs may differentiate into osteoblasts due to continuous exposure to these supplements [57]. To accelerate adipose lineage commitment, we introduced rosiglitazone, a PPARγ agonist that plays a central role in regulating adipogenesis, during the first nine days of culture. As expected, rosiglitazone directed the rapid and robust commitment of hBMSCs to preadipocytes. Subsequently, by maintaining preadipocytes with insulin alone, we continued their expansion and hypertrophy of lipid droplets while reducing the chance for osteogenic differentiation of hBMSCs [58]. This two-stage adipogenic differentiation strategy resulted in the generation of a significant number of mature adipocytes with ~2-fold increased lipid droplet size and ~3-fold increased surface area coverage after 18 days compared to conventional adipogenic differentiation medium.

While adipocyte culture on TCP is useful for accessible and controlled experiments, the lack of essential microenvironmental cues derived from neighboring cells and the surrounding extracellular matrix milieu restricts the predictive power of 2D tissue culture studies [59]. Several 3D adipocyte culture models have been introduced to better recapitulate the tissue complexity found in the body [22,24,25]. These models have validated differences in secretory profiles, bone marrow adipocyte morphology, and metabolic activity compared to 2D TCP. In this study, we utilized ICC hydrogel scaffolds with fully interconnected spherical pore arrays made with synthetic polyacrylamide hydrogel to recapitulate the microarchitecture of native bone marrow. The semitransparent hydrogel matrix permitted monitoring of the spheroid formation of hBMSCs, adipogenic differentiation, and sequestration of doxorubicin by lipid droplets. The effect of doxorubicin sequestration on multiple myeloma post-chemotherapy was significantly different compared to 2D TCP models. Our study revealed two key differences between the 3D adipocyte spheroid and the 2D adipocyte on TCP. First, the diameter of lipid droplets in 3D spheroid culture was ~3-fold smaller than the 2D culture on TCP. Second, the increasing rate of lipid droplets in 3D spheroid culture was slower than in 2D culture, despite the addition of rosiglitazone.

This study has several limitations. First, although in vitro differentiated bone marrow adipocytes sequestered doxorubicin in their lipid droplets, the long-term fate of sequestered doxorubicin was not determined. The sequestered drug may be metabolically degraded or released back into the medium, which could still induce cytotoxic effects on surrounding cells. Second, to better understand and compare in vitro differentiated bone marrow adipocytes to their native counterparts, it is essential to characterize their secretion profiles, responsiveness to lipolytic stimuli, and deposition of a basement membrane-like matrix [60,61]. Future studies could incorporate additional assays to examine these factors, providing a more accurate representation of the in vitro behavior of bone marrow adipocytes. Third, while nuclear staining with DAPI is a simple, inexpensive, and high-throughput method for quantitatively identifying cell death, it cannot differentiate between normal cellular apoptosis and necrosis. Other methods, such as TUNEL or propidium iodide staining, could detect DNA fragmentation and plasma membrane disruption due to cytotoxic stress, hallmarks of necrotic cell death [62]. Additional markers for determining specific cellular states would provide a more comprehensive understanding of the impact of chemotherapy on bone marrow adipocytes. Finally, advancements to these in vitro cultures can be made by integrating a bone organoid model and incorporating other marrow components to better replicate the complexity of bone marrow tissue and improve its predictive power [60].

Our study validated significant doxorubicin uptake in in vitro differentiated bone marrow adipocytes in 2D TCP and 3D bone marrow adipocyte spheroids. Furthermore, the sequestration of doxorubicin significantly reduced the cytotoxicity experienced by hBMSCs in conditioned medium experiments. Direct coculture with multiple myeloma allowed us to conclude that bone marrow adipocyte sequestration of doxorubicin reduced its therapeutic efficacy and cytotoxicity through sequestration in the context of cancer. Interestingly, we also observed gradually decreased doxorubicin fluorescent intensity, particularly in 3D adipocyte spheroids, indicating that the sequestered doxorubicin could be metabolized in adipocytes. Overall, these results substantiate the role and importance of bone marrow adipocytes in developing chemoresistance, thereby promoting the survival and proliferation of residual tumor cells in the bone marrow microenvironment (Figure 6).

## 5. Conclusions

We have developed a 3D in vitro differentiated human bone marrow adipocyte model by incorporating rosiglitazone and marrow-mimicking ICC hydrogel scaffolds. Using the established 3D adipocyte spheroid models, we have demonstrated the sequestration of lipophilic doxorubicin by fat droplets and the consequent modulation of cytotoxicity and efficacy. We envision that in vitro 3D human bone marrow models will be a valuable tool for basic and translational investigations into the bone marrow microenvironment, particularly in the context of chemotherapy and sequestration. The use of this model in future studies may help to understand how microenvironmental chemoresistance develops and identify potent targets in the tumor microenvironments that play a significant role in developing drug resistance [61,62].

## Figures and Tables

**Figure 1 cancers-15-02737-f001:**
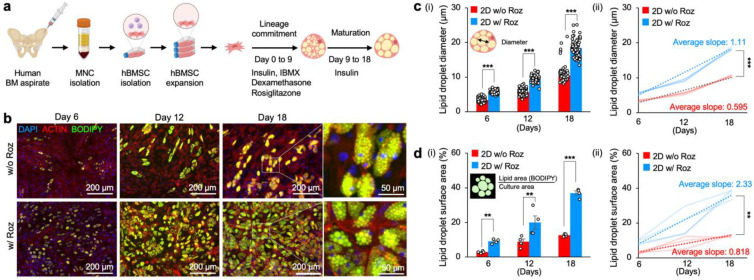
Rosiglitazone significantly accelerates the adipogenesis of primary hBMSCs on 2D TCP. (**a**) Experimental procedure of hBMSC isolation, expansion, and adipogenic differentiation on 2D TCP. (**b**) Representative time-course fluorescence microscopy images of hBMSCs undergoing adipogenic differentiation in vitro with and without rosiglitazone: blue (nucleus, DAPI), red (actin filament, phalloidin), and green (lipid droplet, BODIPY). (**c**(**i**)) Time-course measurements of lipid droplet diameters in adipocytes differentiated with and without rosiglitazone (*n* = 64) and (**c**(**ii**)) quantitative comparison of increasing rates of lipid droplet diameters in adipocytes differentiated with and without rosiglitazone (*n* = 4 independent experiments). (**d**(**i**)) Time-course measurements of normalized surface area covered by lipid droplets (BODIPY) in adipocytes differentiated with and without rosiglitazone (*n* = 4 independent experiments) and (**d**(**ii**)) quantitative comparison of increasing rates of lipid droplet surface areas in adipocytes differentiated with and without rosiglitazone (*n* = 4 independent experiments). The statistical analysis of (**c**(**i**),**d**(**i**)) was performed with a two-way ANOVA followed by a Bonferroni post-test. The statistical analysis of (**c**(**ii**),**d**(**ii**)) was performed with an unpaired, two-tailed Student’s *t*-test. (** *p* < 0.01, *** *p* < 0.001).

**Figure 2 cancers-15-02737-f002:**
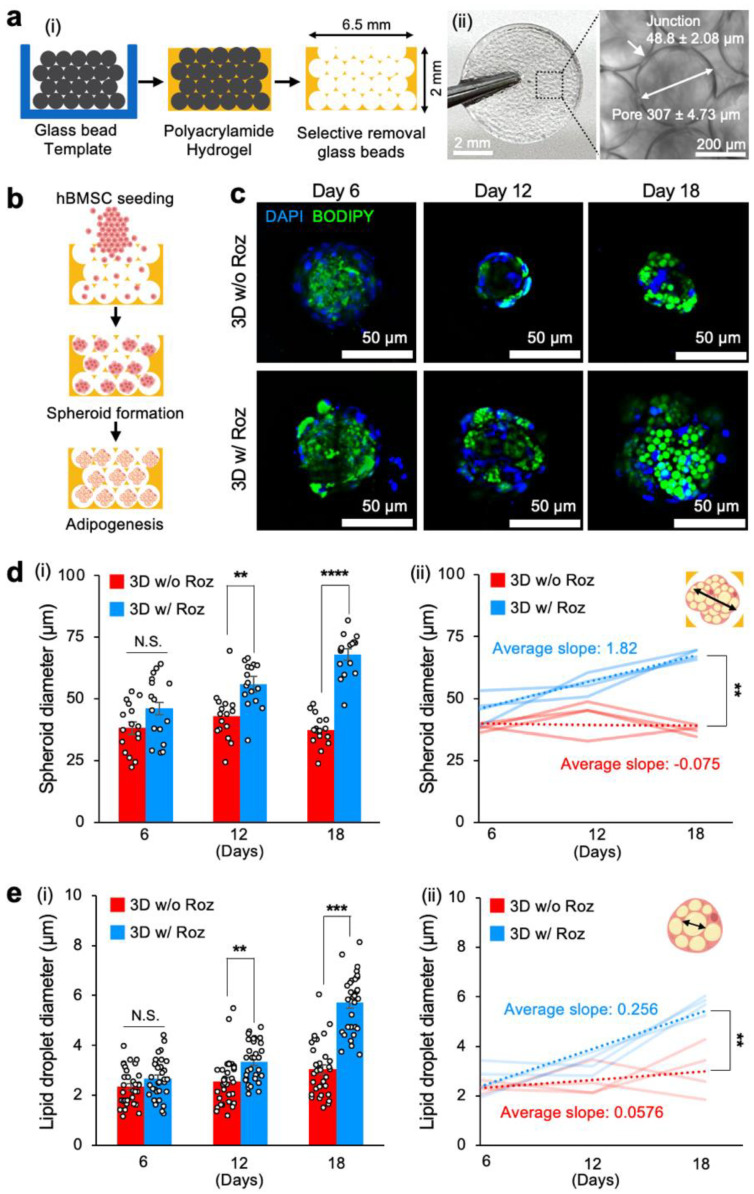
Rosiglitazone significantly accelerates in vitro adipogenic differentiation of hBMSCs in 3D spheroids when cultured in ICC hydrogel scaffolds. (**a**(**i**)) Schematic of microbead template-based ICC hydrogel scaffold fabrication and (**a**(**ii**)) images of a representative porous hydrogel scaffold with standardized spherical pore arrays and interconnecting junctions. (**b**) Experimental procedure to seed hBMSCs into an ICC hydrogel scaffold for spheroid aggregation and subsequent adipogenic differentiation. (**c**) Representative time-course fluorescence microscopy images of hBMSCs undergoing adipogenic differentiation in ICC scaffolds: blue (nucleus, DAPI) and green (lipid droplets, BODIPY). (**d**(**i**)) Time-course measurement of adipocyte spheroid diameters within each pore cavity differentiated with and without rosiglitazone (*n* = 16) and (**d**(**ii**)) quantitative comparison of increasing rates of spheroid diameters (*n* = 4 independent experiments). (**e**(**i**)) Time-course measurements of lipid droplet diameters in adipocyte spheroids differentiated with and without rosiglitazone (*n* = 64) and (**e**(**ii**)) quantitative comparison of increasing rates of lipid droplet diameters (*n* = 4 independent experiments). The statistical analysis of (**d**(**i**),**e**(**i**)) was performed with a two-way ANOVA followed by a Bonferroni post-test. The statistical analysis of (**d**(**ii**),**e**(**ii**)) was performed with an unpaired, two-tailed Student’s *t*-test. (N.S.: not significant, ** *p* < 0.01, *** *p* < 0.001, **** *p* < 0.0001).

**Figure 3 cancers-15-02737-f003:**
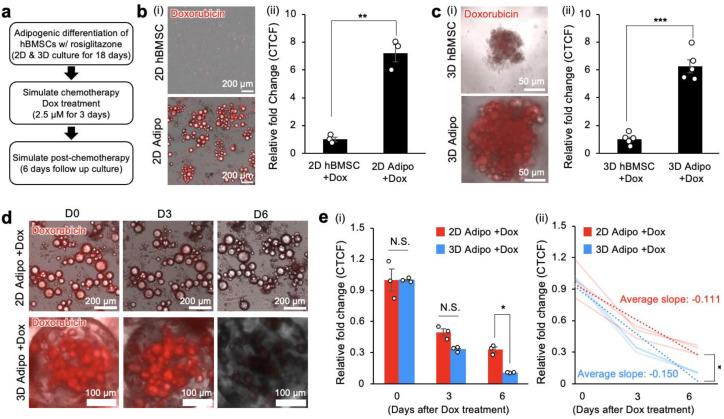
Bone marrow adipocytes in 2D and 3D cultures sequester doxorubicin. (**a**) Experimental procedure and timeline to establish 2D and 3D adipocyte culture models with rosiglitazone, simulate chemotherapy through doxorubicin treatment, and subsequent 6-day post-chemotherapy follow-up. (**b**(**i**)) Representative transmission and red fluorescence (doxorubicin) overlayed images of hBMSCs and adipocytes in 2D TCP at the end of 3 days of doxorubicin treatment, and (**b**(**ii**)) quantitative comparison of relative fold change in CTCF (*n* = 3 independent experiments). (**c**(**i**)) Representative transmission and red fluorescence (doxorubicin) overlayed images of hBMSCs and adipocyte spheroids in 3D hydrogel scaffolds at the end of 3 days of doxorubicin treatment, and (**c**(**ii**)) quantitative comparison of relative fold change in CTCF (*n* = 5). (**d**) Representative time-course transmission and red fluorescence (doxorubicin) overlayed images of 2D TCP and 3D adipocyte spheroids over 6 days post-doxorubicin treatment. (**e**(**i**)) Quantitative measurement of decreasing doxorubicin-derived CTCF fold-change in 2D TCP and 3D adipocytes over 6 days post-doxorubicin treatment (*n* = 3 independent experiments) and (**e**(**ii**)) quantitative comparison of decreasing rates between 2D TCP and 3D adipocyte spheroid CTCF relative fold-change. The statistical analysis of (**b**(**ii**),**c**(**ii**),**e**(**ii**)) was performed with an unpaired, two-tailed Student’s *t*-test. The statistical analysis of (**e**(**i**)) was performed with a two-way ANOVA followed by a Bonferroni post-test. (N.S.: not significant, * *p* < 0.05, ** *p* < 0.01, *** *p* < 0.001).

**Figure 4 cancers-15-02737-f004:**
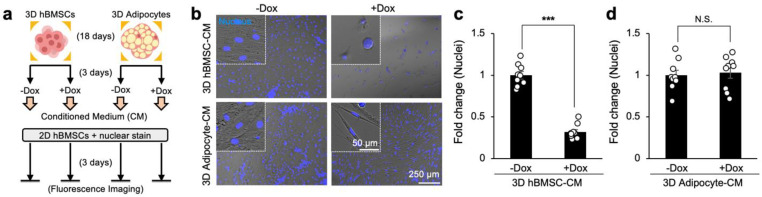
Sequestration of doxorubicin by 3D human adipocytes reduces cytotoxicity in conditioned media to 2D hBMSCs. (**a**) Experimental procedure, groups, and timeline to indirectly determine bone marrow adipocyte sequestration of doxorubicin to determine its contributions to chemoresistance. (**b**) Representative transmission and DAPI fluorescence (Hoechst 33342) overlayed images of naïve 2D hBMSCs on TCP cultured with conditioned media from 3D hBMSC and adipocyte spheroids with or without doxorubicin treatment. (**c**) Quantification of fold-change in the number of nuclei for naïve hBMSCs incubated with conditioned media from 3D hBMSC spheroids with and without doxorubicin (*n* = 9). (**d**) Quantification of fold-change in the number of nuclei for naïve hBMSCs incubated with conditioned media from 3D adipocyte spheroids with and without doxorubicin (*n* = 9). The statistical analysis was performed with an unpaired, two-tailed Student’s *t-*test. (N.S.: not significant, *** *p* < 0.001).

**Figure 5 cancers-15-02737-f005:**
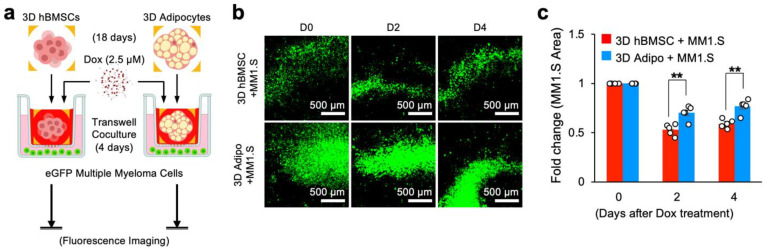
Sequestration of doxorubicin by 3D human bone marrow adipocyte spheroids reduces multiple myeloma cytotoxicity in direct coculture. (**a**) Experimental procedure, groups, and timeline to determine direct adipocyte sequestration of doxorubicin and its contribution towards chemoresistance in a direct Transwell coculture model with human multiple myeloma cells (eGFP-MM1.S). (**b**) Representative time-course GFP fluorescence images of eGFP-MM1.S cells directly cultured with 3D hBMSC and adipocyte spheroids with a Transwell insert (pore diameter: 0.38 µm) following doxorubicin addition for 4 days. (**c**) Quantitative comparison of surface area covered by eGFP-MM1.S cells in direct coculture with 3D hBMSC and adipocyte spheroids with doxorubicin treatment (*n* = 9). The statistical analysis was performed with a two-way ANOVA followed by a Bonferroni post-test. (N.S.: not significant, ** *p* < 0.01).

**Figure 6 cancers-15-02737-f006:**
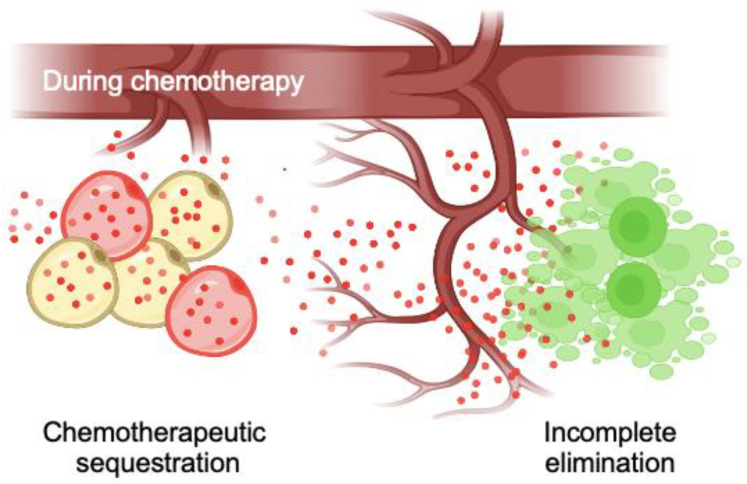
Proposed roles of bone marrow adipocytes in promoting tumor microenvironment-driven chemoresistance. Adipocytes within the bone marrow microenvironment sequester lipophilic chemotherapeutic agents, leading to decreased efficacy and incomplete elimination of tumor cells due to potentially reduced cytotoxicity and chemotherapy effectiveness.

## Data Availability

The authors declare that all data supporting the findings of this study are contained within the paper. Raw data are available upon reasonable request from the corresponding author.

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
