# Peer review of "Bone Marrow Adipocytes Contribute to Tumor Microenvironment-Driven Chemoresistance via Sequestration of Doxorubicin"

_cancers, 2023, doi:10.3390/cancers15102737_

Round 1
Reviewer 1 Report
This study uses 2D and 3D in vitro culture techniques priorly developed to address the issue of chemotherapy sequestration (specifically doxorubicin) by bone marrow adipocytes, and how this impact the targeting of cancer cells. Prior to this, an assay is optimized to improve in vitro differentiation of adipocytes from primary, human bone marrow stroma cells, which was achieved through the addition of rosiglitazone. Using in vitro-differentiated bone marrow adipocytes as monolayers or as spheroids inside bone marrow-like hydrogel-surrounded cavities, and following their fluorescence after the addition of doxorubicin, they demonstrate that adipocytes sequester doxorubicin. This parallels a reduction in the capacity of doxorubicin-containing adipocyte conditioned media to target multiple myeloma cells.
The most interesting aspects of this study are the optimization of a method to study adipocyte-cancer cell crosstalk within a bone-marrow like 3D architecture, and the validation of the method to address a specific question, such as how adipocytes may facilitate the survival of cancer cells by reducing drug concentration.
The methods were well explained and sufficient in their majority for the purpose of the study. The figures are clear, and summarize the results very nicely. Below are a few issues identified that will be important to address to enhance the quality of the manuscript:
1. It is unclear what happens to the doxorubicin that is “sequestered” by adipocytes, as its subcellular localization is not determined. Also, you mentioned that the drug decline over time indicates it is being metabolized. Looking at drug metabolites will be a more accurate way to affirm this.
2. The limitations of the hydrogel bone marrow model used is not properly addressed in the discussion. For instance, are the adipocytes spheroids behaving in a similar way to actual bone marrow adipocytes in terms of traits such as basement membrane-like matrix deposition and other features?
3. You use the term “cytotoxicity” to refer to the reduction in Hoechst 33342 nuclei labelling you observed by IF. Can you affirm it is cytotoxicity you are observing here? It is also important to discuss the limitations of this technique to assess cytotoxicity.
4. It will more accurate to refer to the adipocytes used as in vitro-differentiated bone marrow adipocytes instead of bone marrow adipocytes
4. The conclusion reads more like Future Directions that could be incorporated into the discussion. It will be better to limit the Conclusion section to what can actually be concluded from the data in this study.
5. Finally, there are typing mistakes, missing parts, and other errors throughout the manuscript, for which careful proof-reading is recommended. Here are some examples:
Line 150: delete the word “tissue” before 37oC.
Line 219: the Day 12 diameter of lipid droplets formed without rosiglitazone is missing.
Line 220: at the end of the line, delete “and” after “mean diameter”.
Line 223: at the beginning of the line, change “higher” to “faster”.
Lines 229-230: there is no difference in the lipid droplet diameter with and without rosiglitazone.
Line 316: delete “for”
Line 465: “introducing” might not be the correct word here.
Just typing mistakes. Careful proof-reading by authors highly recommended.
Author Response
Reviewer 1
This study uses 2D and 3D in vitro culture techniques priorly developed to address the issue of chemotherapy sequestration (specifically doxorubicin) by bone marrow adipocytes, and how this impact the targeting of cancer cells. Prior to this, an assay is optimized to improve in vitro differentiation of adipocytes from primary, human bone marrow stroma cells, which was achieved through the addition of rosiglitazone. Using in vitro-differentiated bone marrow adipocytes as monolayers or as spheroids inside bone marrow-like hydrogel-surrounded cavities, and following their fluorescence after the addition of doxorubicin, they demonstrate that adipocytes sequester doxorubicin. This parallels a reduction in the capacity of doxorubicin-containing adipocyte conditioned media to target multiple myeloma cells.
The most interesting aspects of this study are the optimization of a method to study adipocyte-cancer cell crosstalk within a bone-marrow like 3D architecture, and the validation of the method to address a specific question, such as how adipocytes may facilitate the survival of cancer cells by reducing drug concentration.
The methods were well explained and sufficient in their majority for the purpose of the study. The figures are clear and summarize the results very nicely. Below are a few issues identified that will be important to address to enhance the quality of the manuscript:
Overall response: We sincerely thank Reviewer 1 for investing valuable time and effort into thoroughly reviewing our manuscript and providing constructive feedback. We have considered the points raised in this revised version and made significant improvements accordingly. We hope Reviewer 1 looks favorably at our revised manuscript.
- It is unclear what happens to the doxorubicin that is “sequestered” by adipocytes, as its subcellular localization is not determined. Also, you mentioned that the drug decline over time indicates it is being metabolized. Looking at drug metabolites will be a more accurate way to affirm this.
Response: Thank you for the comment. We present in this study a compelling and reproducible observation regarding the sequestration of doxorubicin by lipid droplets, which gradually disappears due to lack of autofluorescence. However, the fate of sequestered doxorubicin within lipid droplets remains unanswered. Our imaging studies did not reveal accumulation of fluorescence in specific subcellular organelles, and post-doxorubicin sequestration bone marrow adipocyte cultures did not exhibit significant cytotoxicity. Therefore, we hypothesized that doxorubicin may undergo a chemical transformation or metabolized, thereby losing its intrinsic fluorescence through metabolism. We plan to investigate and further characterize this process in our future studies. In this revised version, we have discussed the fate of post-sequestered doxorubicin, as outlined below:
“This study has several limitations. First, although in vitro differentiated bone marrow adipocytes sequestered doxorubicin in their lipid droplets, the long-term fate of sequestered doxorubicin was not determined. The sequestered drug may be metabolically degraded or released back into the medium, which could still induce cytotoxic effects on surrounding cells.”
- The limitations of the hydrogel bone marrow model used is not properly addressed in the discussion. For instance, are the adipocytes spheroids behaving in a similar way to actual bone marrow adipocytes in terms of traits such as basement membrane-like matrix deposition and other features?
Response: Thank you for the comment. Our hydrogel scaffold enables recapitulation of the bone marrow microenvironment by recapitulating geometrical and biophysical properties but is limited to reproducing different biochemical, mechanical (shear, compression), and multicellular complexity present in native tissue. In this revision, we addressed this concern below:
“Second, to better understand and compare in vitro differentiated bone marrow adipocytes to their native counterparts, it is essential to characterize their secretion profiles, responsiveness to lipolytic stimuli, and deposition of a basement membrane-like matrix. Future studies could incorporate additional assays to examine these factors, providing a more accurate representation of the in vitro behaviors of bone marrow adipocytes.”
- You use the term “cytotoxicity” to refer to the reduction in Hoechst 33342 nuclei labelling you observed by IF. Can you affirm it is cytotoxicity you are observing here? It is also important to discuss the limitations of this technique to assess cytotoxicity.
Response: In this study, we characterized cytotoxicity by examining cellularity changes before and after doxorubicin treatment using nuclei staining for live cells. We included a control group without doxorubicin treatment, where the number of nuclei (DAPI) staining remained stable.
Based on these experiments, we quantitatively assessed the "cytotoxicity" induced by doxorubicin solely. However, we acknowledge that our assessment lacked the characterization of specific biomarkers for cytotoxicity and did not distinguish between necrosis and apoptosis. We recognize that a more detailed evaluation of cellular stress and cytotoxic mechanisms is crucial.
In this revised manuscript, we have discussed the limitations of the presented cytotoxic assay, which relies solely on nuclei staining count. We emphasize the importance of utilizing more specific functional assays to complement our findings. The following statements have been included in the manuscript to address this limitation:
“Third, while nuclear staining with DAPI is a simple, inexpensive, and high-throughput method for quantitatively identifying cell death, it cannot differentiate between normal cellular apoptosis and necrosis. Other methods, such as TUNEL or propidium iodide staining, could detect DNA fragmentation and plasma membrane disruption due to cytotoxic stress, hallmarks of necrotic cell death. Additional markers for determining specific cellular states would provide a more comprehensive understanding of the impact of chemotherapy on bone marrow adipocytes.”
- It will be more accurate to refer to the adipocytes used asin vitro-differentiated bone marrow adipocytes instead of bone marrow adipocytes
Response: Thank you for the comment. This is a more accurate description and potentially avoids any misreading. Therefore, in this revision, we have changed “bone marrow adipocytes” to in vitro-differentiated bone marrow adipocytes”.
- The conclusion reads more like Future Directions that could be incorporated into the discussion. It will be better to limit the Conclusionsection to what can actually be concluded from the data in this study.
Response: Thank you for your careful reading and suggestion. We agree with the reviewer’s suggestion. Accordingly, we updated our conclusion in this revision as below.
“We have developed a 3D in-vitro differentiated human bone marrow adipocyte model by incorporating rosiglitazone and marrow-mimicking ICC hydrogel scaffolds. Using the established 3D adipocyte spheroid models, we have demonstrated the sequestration of lipophilic doxorubicin by fat droplets and the consequent modulation of cytotoxicity and efficacy. We envision that in vitro 3D human bone marrow models will be a valuable tool for basic and translational investigation into the bone marrow microenvironment, particularly in the context of chemotherapy and sequestration. The use of this model in future studies may help to understand how microenvironmental chemoresistance develops and identify potent targets in the tumor microenvironments that play a significant role in developing drug resistance.”
- Finally, there are typing mistakes, missing parts, and other errors throughout the manuscript, for which careful proof-reading is recommended. Here are some examples:
Line 150: delete the word “tissue” before 37oC.
Line 219: the Day 12 diameter of lipid droplets formed without rosiglitazone is missing.
Line 220: at the end of the line, delete “and” after “mean diameter”.
Line 223: at the beginning of the line, change “higher” to “faster”.
Lines 229-230: there is no difference in the lipid droplet diameter with and without rosiglitazone.
Line 316: delete “for”
Line 465: “introducing” might not be the correct word here.
Response: We express our sincere gratitude for your careful review of our manuscript and for bringing specific issues to our attention. In this revised version, we have substantially improved the writing style, added clarity, and removed grammatical and typographical mistakes. We have also conducted a thorough proofreading of the entire manuscript and addressed these issues.

Reviewer 2 Report
The authors have done nice work by studying the bone marrow adipocytes contributing to tumor microenvironment driven chemoresistance. Authors have developed a protocol for the rapid and robust differentiation of human bone marrow stromal cells (hBM- SCs) into mature adipocytes. And how mature adipocytes sequester chemotherapy drugs in lipid droplets, which reduces the cytotoxicity. The topic is relevant in the field. Such studies will accelerate findings to improve traditional cancer treatment approaches taken into account the microenvironmental chemoresistance. This study will help to explore the ways for increased efficacy and safety of conventional cancer treatment by avoiding overtreatment, and reducing toxic side effects. In methods part co-culture assay of multiple myeloma cells and hBMSC/adipocyte could have been monitored by using the live imaging or Incucyte. That will give better visualization and kinetics. The references are appropriate. The conclusions are consistent with the evidence and arguments presented and address the main question posed. Although the authors have done good work the manuscript needs improvement in the English and the scientific language.
Minor Revision is recommended.
English and scientific language needs to be checked.
Author Response
Reviewer 2
The authors have done nice work by studying the bone marrow adipocytes contributing to tumor microenvironment driven chemoresistance. Authors have developed a protocol for the rapid and robust differentiation of human bone marrow stromal cells (hBMSCs) into mature adipocytes. And how mature adipocytes sequester chemotherapy drugs in lipid droplets, which reduces the cytotoxicity. The topic is relevant in the field. Such studies will accelerate findings to improve traditional cancer treatment approaches taken into account the microenvironmental chemoresistance. This study will help to explore the ways for increased efficacy and safety of conventional cancer treatment by avoiding overtreatment, and reducing toxic side effects. In methods part co-culture assay of multiple myeloma cells and hBMSC/adipocyte could have been monitored by using the live imaging or Incucyte. That will give better visualization and kinetics. The references are appropriate. The conclusions are consistent with the evidence and arguments presented and address the main question posed. Although the authors have done good work the manuscript needs improvement in the English and the scientific language.
Minor Revision is recommended.
Response: We appreciate Reviewer 2’s time and effort in reviewing our manuscript. In this revision, we have significantly improved our manuscript to make it clear and easy to read. We have also carefully edited the manuscript for English grammar and typos.
We appreciate the insightful suggestion of time-course microscopy imaging of co-culture experiments. Currently, we are limited in conducting this highly valuable experiment. We plan to perform these experiments in future studies.

Reviewer 3 Report
I have several comments in the submitted paper as follows:
1. Nothing truly unique in its current state. Because of the lack of a novel, the current submission looks to be a replication or modified work. The authors must describe their novel in detail. This work should be rejected owing to a major issue.
2. The work, novelty, and limitations of similar prior studies must be explained in the introduction section to highlight the literature gaps that the current work aims to fill.
3. The authors needs to explain potential further study performing in silico/computational simulation. It bring several advantages compared to in vivo and in vitro such as lower cost and faster results. In silico also would become preliminary investigation before performing in vitro and/or in vivo study. Please include the relevant reference to support the explanation as follows: In Silico Contact Pressure of Metal-on-Metal Total Hip Implant with Different Materials Subjected to Gait Loading. Metals (Basel). 2022;12: 1241. doi:10.3390/met12081241
4. Please include the limitation of the present submission, it is missing.
-
Author Response
Reviewer 3
- Nothing truly unique in its current state. Because of the lack of a novel, the current submission looks to be a replication or modified work. The authors must describe their novel in detail. This work should be rejected owing to a major issue.
Response: We improved the manuscript to clarify the novelty and significance of the current work. We believe the current work will make a meaningful contribution to the tumor microenvironment and chemoresistance fields.
- The work, novelty, and limitations of similar prior studies must be explained in the introduction section to highlight the literature gaps that the current work aims to fill.
Response: We have significantly improved this manuscript to enhance the clarity and readability of the novelty and limitations sections.
- The authors needs to explain potential further study performing in silico/computational simulation. It bring several advantages compared to in vivo and in vitro such as lower cost and faster results. In silico also would become preliminary investigation before performing in vitro and/or in vivo study. Please include the relevant reference to support the explanation as follows: In Silico Contact Pressure of Metal-on-Metal Total Hip Implant with Different Materials Subjected to Gait Loading. Metals (Basel). 2022;12: 1241. doi:10.3390/met12081241.
Response: We believe this comment is not relevant to the current manuscript. The suggested reference is unrelated to this manuscript and does not fit in the context and scope of this paper.
- Please include the limitation of the present submission, it is missing.
Response: We included limitations and potential future directions in this revision.
